# Impact of COVID-19 on Health-Related Quality of Life: A Longitudinal Study in a Spanish Clinical Sample

**DOI:** 10.3390/ijerph191610421

**Published:** 2022-08-21

**Authors:** Irene Rodríguez-Galán, Natalia Albaladejo-Blázquez, Nicolás Ruiz-Robledillo, José Francisco Pascual-Lledó, Rosario Ferrer-Cascales, Joan Gil-Carbonell

**Affiliations:** 1Pneumology Department, Alicante General University Hospital—Alicante Institute of Health and Biomedical Research (ISABIAL), 03010 Alicante, Spain; 2Department of Health Psychology, University of Alicante, 03690 Alicante, Spain

**Keywords:** COVID-19, health-related quality of life, sequelae, life quality, pneumonia

## Abstract

SARS-CoV-2 respiratory infection and the course of its sequelae remain to be defined. The aim of this study is to analyze health status and Health-Related Quality of Life (HRQoL) in a Spanish sample of survivors of coronavirus disease 2019 (COVID-19) pneumonia. Methods: Prospective observational study of patients who survived SARS-CoV-2 pneumonia, between February 2020 and May 2020, with systematic evaluation at 3 and 12 months after the onset of the disease. The data were obtained by reviewing the clinical history and performing a physical examination, a chest X-ray, and a pulmonary function test on the patients. Additionally, the SF-36 questionnaire was administered for the HRQoL study. Results: In total, 130 patients aged 55.9 ± 15.9 years were included. Dyspnea (36.9%) and asthenia (36.2%) were the most frequent persistent symptoms. Fibrotic pulmonary changes were detected in 20.8% of the participants. Compared to the general population, significant deterioration was detected in all domains of the SF-36 questionnaire at 3 and 12 months post-COVID-19 infection. The greatest differences were in the physical role (RF) and in the emotional role (RE). Conclusions: COVID-19 pneumonia causes a long-term deterioration in HRQoL compared to the general population. Over time, a trend toward improvement is detected in most domains of the SF-36.

## 1. Introduction

The appearance of SARS-CoV-2 coronavirus respiratory infection in December 2019, followed by the global outbreak in 2020 and its permanence today, with more than 290 million diagnosed cases [1] has become a global health problem.

The most severe manifestation of the disease is pneumonia evidenced by the appearance of fever, cough, dyspnea, and bilateral pulmonary opacities on chest X-rays. In a study of over 70,000 cases, 81% had mild forms (with or without mild pneumonia), 14% were moderate (pneumonia with hypoxemia), and 5% exhibited severe symptoms (respiratory failure requiring mechanical ventilation, shock, or multiorgan failure) [2].

Overall mortality in large groups of patients has been recorded at 2.3% [2], which rises to 60% in patients with severe forms requiring admission to the Intensive Care Unit (ICU) [3].

Recovery time is variable but is generally about two weeks in mild forms and can last up to three months in more severe forms. In 10% of infected patients, symptoms persist beyond the expected time, which is known as long COVID or persistent COVID [4].

The impact on HRQoL as a consequence of respiratory infection is a subject of study nowadays. Several systematic reviews and meta-analyses on the effects resulting from infection and its impact on quality of life have recently been published [5,6,7,8].

Narayan Poudel Ak et al. [5] analyzed 12 publications (out of 1276), that followed patients up to 12 weeks after hospital discharge. Among the tools used to study HRQoL, five publications used the SF-36 questionnaire, five the EQ-5D-5L questionnaire, one the PROMIS tool and two used specific lung disease questionnaires (St George’s Respiratory Questionnaire, Clinical COPD). Of those using the SF-36 questionnaire, only one of them compared data from the acute phase (<4 weeks) and stable phase (>4 weeks) and a worsening in quality of life is observed in both acute COVID and stable COVID (12 weeks). This impairment is more prevalent in the acute phase, in women, in older age, in patients with more severe disease, and in patients from low-income countries. Furthermore, in the acute phase, a worse clinical score is observed in the physical component, which is reversed in the stable phase, with worse scores in the mental component.

Sanchez-Ramirez et al. [6] examined four studies from a total of 5054 articles to assess HRQoL four months after infection. Patients with severe forms of the disease had greater impairment in quality of life according to the EQ-5D5L questionnaire. However, patients with mild disease reported significantly worse health in the SF-36 questionnaire and in the subdomains of the Nijmegen Clinical Screening Instrument [7].

Malik et al. [8] analyzed 12 studies from a total of 4828 with the aim of describing the prevalence of poor quality of life and analyzing the effect of persistent symptoms and ICU stay. The studies followed patients between one and six months after hospital discharge. They concluded that the majority of patients who had recovered after COVID-19 disease had a poor quality of life, with a prevalence of poor quality of life of 59%, as measured by EQ-VAS. Among the individual factors, the EQ-5Q-5L questionnaire showed that 41.5% have pain/discomfort, 37.5% anxiety/depression, 36% mobility problems, 28% problems with usual activities and 8% self-care problems. Meta-regression analysis showed that poor quality of life is significantly higher in patients who have required admission to the ICU and in those with fatigue as a persistent symptom.

Regarding other similar diseases, according to the data available from the study by Hui et al. [9] on the effects of the SARS-CoV epidemic in 2003, one year after hospitalization about 25% of the patients had residual fibrotic lung lesions and decreased pulmonary diffusion. Additionally, patients exhibited a deterioration in quality of life compared to the general population and a significant decrease in most domains of the SF-36 questionnaire.

Taking into account the previously mentioned studies, a deterioration in health status has been proved in COVID-19-infected patients, in the acute phase (4 weeks), sub-acute phase (4–12 weeks), and stable phase (12 weeks–6 months). However, more research is needed on the long-term health impact of the SARS-CoV-2 infection. It is necessary to characterize which domains are most affected and whether there is improvement over time. This would optimize the implementation of tailored multidisciplinary rehabilitation programs (physical, psychological, and psychiatric) which, at present, constitute the fundamental pillar in the management of the after-effects derived from COVID-19 [10,11].

Therefore, the objective of this study was to evaluate the impact of COVID-19 disease on HRQoL and its evolution over time (at 3 and 12 months after the infection) in a Spanish sample of survivors of bilateral SARS-CoV-2 pneumonia.

## 2. Materials and Methods

### 2.1. Study Design and Participants

This prospective observational study follows patients who suffered from SARS-CoV-2-related pneumonia between February and May 2020. The inclusion criteria were a confirmed SARS-CoV-2 infection via Real Time Reverse transcription polymerase chain reaction (RT-PCR) and a positive diagnostic for pneumonia. Samples for the RT-PCR test were collected through nasopharyngeal aspirate or sampling from the lower respiratory tract. The pneumonia diagnostic was done by assessing the patient’s symptoms and performing a pulmonary X-ray. All patients were older than 18 years of age.

### 2.2. Procedure

Patients were evaluated between 10 and 14 weeks after hospital discharge for SARS-CoV-2 related pneumonia. Demographic, radiological and spirometric data were collected at the first visit. The SF-36 questionnaire was self-administered at three months, and then another one was administered telematically (via e-mail or telephone call) at 12 months. The reference population values for the Spanish version of the questionnaire were obtained from Alonso et al. [12].

Between February 2020 and May 2020, 305 patients were discharged for SARS-CoV-2-related pneumonia. Of these, 299 (98.0%) attended the proposed 3-month follow-up and 130 (42.6%) completed it. The reasons for patient exclusion during the study are shown in Figure 1.

The study was performed following the standard clinical practice recommendations included in the consensus document of the Spanish Society of Pneumology and Thoracic Surgery (SEPAR) for post-COVID-19 clinical follow-up [13] and has the approval of the Ethics Committee for Drug Research of the Alicante Health Department (Favorable Opinion PI2021-090 (ISABIAL 2021-0145).

### 2.3. Measures

#### 2.3.1. Sociodemographic and Clinical Data

The data were obtained by reviewing the clinical history and performing a physical examination, a chest X-ray, and a pulmonary function test on the patients. Additionally, the participants completed the SF-36 health questionnaire. The aspects researched were smoking habits, comorbidities (arterial hypertension, diabetes mellitus, dyslipidemia or chronic respiratory diseases), signs and symptoms of cough, dyspnea, asthenia, myalgias, arthralgias, anosmia, ageusia, memory loss, skin alterations, headache, and visual loss. Additionally, a physical examination (including oximetry) was performed. The severity of lesions observed on the pulmonary X-rays was assessed according to the modified RALE scale [14,15]. The spirometry was analyzed following the guidelines by American Thoracic Society (ATS) and European Respiratory Society (ERS).

#### 2.3.2. SF-36 Questionnaire

To evaluate HRQOL, the authors used the SF-36 health questionnaire, designed by Ware et al. and adapted to Spanish by Alonso et al. [16].

The questionnaire is a generic HRQoL measure, with a completion time of between five and 10 min, consisting of 36 scorable items, and covering 8 aspects: physical function (PF), physical role (RP), bodily pain (BP), general health (GH), vitality (VT), social function (SF), emotional role (RE), and mental health (MH). The higher the score, the better the health status: the score range is from 0 (worst score) to 100 (best score). It also allows the calculation of a health transition item and two summary scores: the physical summary component (PCS) and the mental summary component (MCS). The Cronbach’s alpha internal consistency coefficient of the questionnaire exceeds the minimum recommended value for group comparisons (Cronbach’s α = 0.7) for all scales except the SF [17]. The RP, PF and RE scales have the best reliability results, and on most occasions, they exceed the value of 0.90, the limit recommended for individual comparisons [17].

### 2.4. Data Analysis

Qualitative variables are described as absolute frequency and percentage, and quantitative variables are described as mean, standard deviation, and range. Differences between general and clinical population in HRQoL domains were evaluated employing independent T-Tests analysis. For purposes of comparison between our data and the values described for the general Spanish population [12], it was taken into account that the distribution of age and sex in our series was similar to that of the general reference population. To analyze differences at 3 and 6 months after the infection a paired samples *t*-test was performed. To assess the size of the effect, Cohen’s d was used, and it was classified as small (d = 0.2–0.3), medium (d = 0.5–0.8) and large (d > 0.8) [18]. The authors used SPSS for windows, version 24.0.0.0 (IBM Corp. Released 2016. IBM SPSS Statistics for Windows, Version 24.0. IBM Corp., Armonk, NY, USA) and MedCalc Statistical Software version 19.2.6 (MedCalc Software bv, Ostend, Belgium; 2020). The limit of statistical significance was set at ≤0.05.

## 3. Results

### 3.1. General Characteristics

The study evaluated a total of 130 patients surviving SARS-CoV-2 pneumonia. The group consisted of 63 males (48.5%) and 67 females (51.5%); with a mean (standard deviation) age of 55.9 (15.9) years.

Demographic characteristics, comorbidities, and symptoms at 12 weeks are shown in Table 1. Age, Charlson index, chest X-ray score according to the modified RALE scale, and spirometric values are shown in Table 2.

The most frequently observed persistent symptoms were dyspnea, asthenia, anosmia-ageusia and cough. Persistent radiological abnormalities were detected in 20.8% of the participants. Respiratory function tests were normal.

### 3.2. Differences in HRQoL between Patients with Post-COVID-19 Pneumonia and General Population

The results of the scores of the different domains of the SF-36 questionnaire, at baseline and at the end of follow-up are shown in Table 3.

Compared to the general population, at 3 months, patients who survived SARS-CoV-2 pneumonia had lower scores in all domains of the SF-36 questionnaire. Large differences were detected in PR (d = 1.1) and SR (d = 1.0). At 12 months, these differences persisted with worse scores in all questionnaire domains except mental health.

### 3.3. Evolution in HRQoL Scores in Patients with Post-COVID-19 Pneumonia at 3 and 12 Months after the Contagion

The results of the analysis of the evolution of the participants one year after hospital discharge are shown in Table 4 and Figure 2. Although a trend towards improvement was detected in most domains of the SF-36 questionnaire, the effect size was practically null for the RP, VT, SF, RE, MH, and PF domains, although for the PF domain, the difference in scores at 3 and 6 months showed a relevant increase. The BP and GH domains also showed improvement over the testing period, although the effect size could be described as small. Finally, regarding the summary scores, a decrease in PCS was found over time, with a small effect size.

## 4. Discussion

This study shows that patients who survived SARS-CoV-2 pneumonia have a worse quality of life than the general population.

At the 3-month follow-up, lower scores are detected in all domains of the SF-36 questionnaire. Studies conducted at six weeks [19] and three months [20] after hospital discharge show similar results. They also describe significant impairment in most domains of the questionnaire, with particularly low scores in functioning and physical role. In our study, the most affected domains were physical role and emotional role, with mean scores of 38.8 and 50.7, respectively.

At the 12-month follow-up, the worsening in HRQoL persisted compared to the general population, with significantly lower scores in all domains except mental health. Aranda et al. [21] examine survivors of severe COVID-19 pneumonia with adult respiratory distress syndrome at 8 months after hospital discharge and also observe a steady deterioration in all domains of the questionnaire.

In this study, we performed an analysis of the evolution of HRQOL in the short (3 months) and long term (12 months). A trend towards improvement is detected in most domains of the SF-36 questionnaire except for FP, BD and GH, where worsening is observed. Published articles on the follow-up of patients with COVID-19 have been based on shorter follow-up periods so very little data are available after one year to contrast these results.

Eberst et al. [22] study survivors of SARS-CoV-2 pneumonia who developed severe forms of the disease and required intensive care unit admission. The only domain in which they detect improvement over the time of 3, 6 and 12 months is the physical role.

Approximately 80% of those infected with SARS-CoV-2 have mild forms of the disease. However, there is a non-negligible percentage of patients with moderate or severe forms, with bilateral pneumonia and who develop multiple complications, the most serious and usually life-threatening of which is adult respiratory distress syndrome. Cardiac (acute myocardial damage, arrhythmias, cardiomyopathy), neurological (headache, dizziness, altered consciousness, cerebrovascular accidents), renal (insufficiency) and haematological (coagulation disorders) involvement, among others, has also been observed.

All this contributes to the development of pulmonary after-effects with persistent radiological and functional abnormalities. A recent study with data up to 1 year reveals that despite the improvement of symptoms, radiological and 1/3 respiratory function abnormalities persist in 1/4 of the cases [23]. However, there is also evidence of Parkinson’s disease and diabetes mellitus in previously healthy people [24] and psychiatric and psychological consequences such as depression and anxiety. Whether the latter are secondary to the inflammatory process of the virus, to post-traumatic stress or to the effects of the treatments has not been clarified [25].

In addition, according to the latest developments in the post-COVID-19 (long COVID) condition, multiple persistent symptoms such as fatigue, headache, attention deficit disorder, dyspnea, cough, anosmia, ageusia, etc., have been described, which are not related to the severity of the disease. Many post-COVID-19 patients report increased suffering due to the non-attention to these symptoms by their physicians (primary care, pulmonology) [26] and up to twice the risk of being diagnosed with a psychiatric disorder after diagnosis of COVID-19 has been reported [27].

In our study, the detected deterioration in physical function, bodily pain and general health could be explained by both the after-effects of the infection and the post-COVID-19 condition. Future studies should analyze new explanatory mechanisms for these results, as well as specific patient profiles with certain clinical characteristics that may be influencing these results.

This study has some limitations. Only 42.6% of all patients admitted for COVID-19 pneumonia completed the questionnaire at 12 months. The sample size may have contributed to the lack of significance in long-term comparisons. Moreover, the selective dropout of participants should be considered, mainly due to lack of motivation, which has generated that some patients who continue with follow-up were those who presented the worst evolution of the disease and needed treatment. However, heterogeneous results have been obtained in relation to the evolution of the quality of life, improving in some dimensions and worsening in others. Furthermore, it was not possible to specifically match by sex and age the comparison groups (general population and clinical sample) due to the data from general population was extracted from a previously published paper. This fact could limit the generalization of the obtained results and entails taking with caution the results obtained. Future studies should address this issue including a control group from the general population specifically matched by these sociodemographic dimensions to analyze the possible differences in HRQoL.

Among the strengths of the study is the follow-up, as there are very few studies available that assess HRQoL over such a long term in survivors of COVID-19 pneumonia.

## 5. Conclusions

Our study confirms that COVID-19 pneumonia causes a deterioration in HRQoL as measured by the SF-36 questionnaire and this affectation can persist over time. The main clinical practice guidelines for the follow-up of post-COVID-19 patients (NICE, WHO) [10,11] with persistent symptoms and sequelae at different levels recommend, with a high degree of evidence, the support of patients with multidisciplinary rehabilitation. It includes giving advice and information on self-management to people with ongoing symptomatic COVID-19 or post-COVID-19 syndrome and use a multidisciplinary approach to guide rehabilitation, including physical, psychological and psychiatric aspects of management. Particular attention should be given to vulnerable people, for example older people and people with complex needs. Additional support may include short-term care packages, advance care planning, and support with social isolation, loneliness, and bereavement, if appropriate.

Our results suggest that such care should be maintained in the long term to minimize the overall impact of the disease on the patient’s overall wellbeing and lifestyle. We consider that more evidence and research is needed to understand the impact of the disease on quality of life in order to design more strategies to improve post-COVID-19 patient care.

## Figures and Tables

**Figure 1 ijerph-19-10421-f001:**
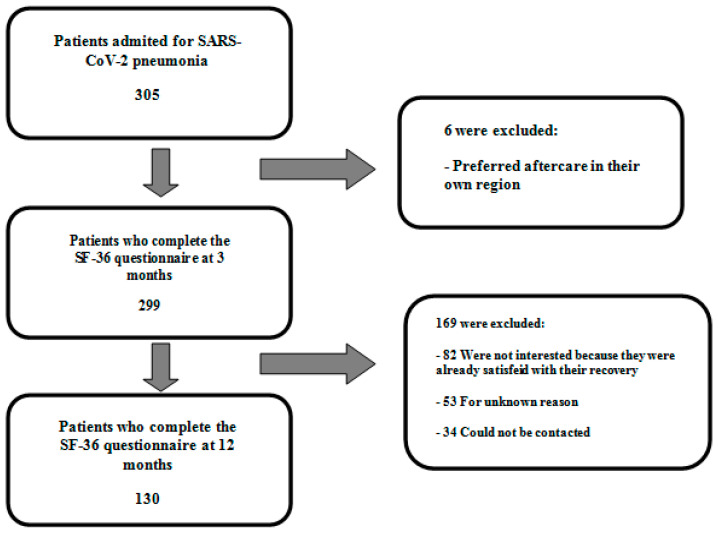
Flow chart of patient inclusion.

**Figure 2 ijerph-19-10421-f002:**
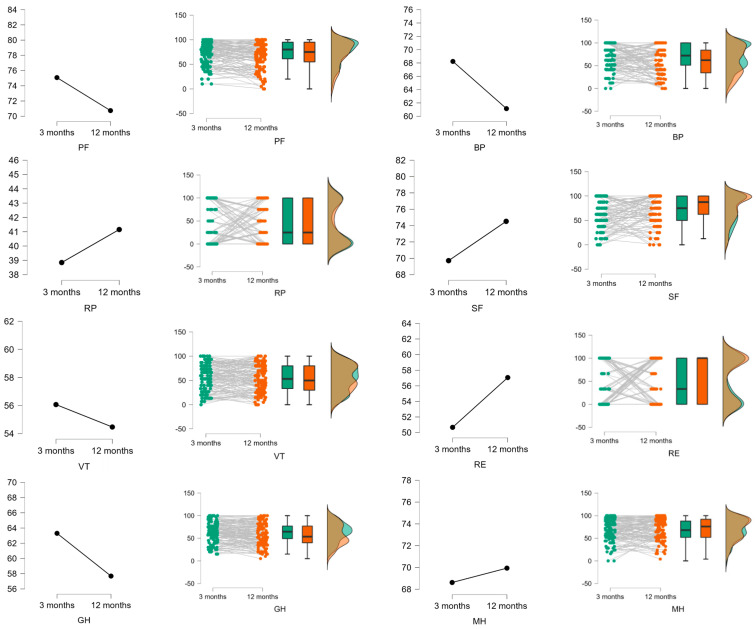
The differences between 3 and 12 months. In each figure, on the ordinate axis, the mean of the domain scores is indicated, and on the abscissa axis, the difference between 3 and 12 months. PF: Physical functioning. BP: Bodily pain. RP: Physical role. SF: Social functioning. VT: Vitality. RE: Emotional role. GH: General health. MH: Mental health. PCS: Physical component summary score. MCS: Mental component summary score.

**Table 1 ijerph-19-10421-t001:** Characteristics of enrolled patients. Qualitative variables.

Characteristics	Total Patients (N = 305) Frequency (%)	Total Enrolled Patients (N = 130) Frequency (%)
Male	157 (51.5%)	63 (48.5%)
Female	148 (48.5%)	67 (51.5%)
Hypertension	104 (34.1%)	50 (38.5%)
Diabetes Mellitus	34 (11.1%)	13 (10.0%)
Pulmonary disease ^1^	50 (16.4%)	20 (15.4%)
Immunosuppression	13 (4.3%)	7 (5.4%)
Obesity	88 (28.9%)	37 (28.5%)
Smoking history	134 (43.9%)	28 (21.5%)
Cough	66 (21.6%)	28 (21.5%)
Dyspnea	100 (32.8%)	48 (36.9%)
Asthenia	102 (33.4%)	47 (36.2%)
Myalgias-arthralgias	56 (18.4%)	25 (19.2%)
Anosmia-dysgeusia	59 (19.3%)	31 (23.8%)
Gastrointestinal disorders	33 (10.8%)	17 (13.1%)
Memory loss	44 (14.4%)	14 (10.8%)
Dermatological disorders	25 (8.2%)	12 (9.2%)
Headache	55 (18.0%)	20 (15.4%)
Visual loss	15 (4.9%)	7 (5.4%)
Back low pain	11 (3.6%)	6 (4.6%)
Crackles	18 (5.9%)	12 (9.2%)
Post-COVID-19 sequels	31 (10.2%)	27 (20.8%)

^1^ Pulmonary disease: chronic obstructive pulmonary disease, asthma, other. In each box, the number of cases is indicated and the percentage in brackets.

**Table 2 ijerph-19-10421-t002:** Characteristics of enrolled patients. Quantitative variables.

Characteristics	Total Patients (N = 305) Mean (SD) (Range)	Total Enrolled Patients (N = 130) Mean (SD) (Range)
Age, years	54.7 (15.9)	55.9 (15.9)
	(18.0–89.0)	(18.0–88.0)
Charlson Index	1.9 (2.1)	1.9 (1.8)
	(0.0–11.0)	(0.0–9.0)
Modified RALE score	0.8 (1.5)	1.1 (1.8)
	(0.0–9.0)	(0.0–9.0)
Pulmonary function		
FVC, mL	4027.5 (1177.2)	3836.1 (1071.1)
	(1450.0–7410.0)	(1550.0–7270.0)
FVC, %	111.6 (18.3)	109.8 (17.2)
	(64.7–161.1)	(64.9–150.6)
FEV1, mL	3104.3 (983.9)	2989.2 (897.6)
	(101.3–5770.0)	(940.0–5530.0)
FEV1, %	107.1 (20.1)	106.1 (18.8)
	(38.7–152.6)	(50.2–147.8)
FEV1/FVC %	77.0 (0.1)	77.8 (8.9)
	(45.0–122.0)	(54.0–122.0)

In each box, in the first row, the mean is indicated and the standard deviation in brackets; and in the second row, the range in brackets. FVC: Forced vital capacity. FEV1: Forced expiratory volume in one second. %: % of predicted, except for FEV1/FVC rate.

**Table 3 ijerph-19-10421-t003:** Values of the scores of the domains of the SF-36 questionnaire in the Spanish population (reference) and in the present study.

SF-36 Domains	General Population (N = 9151)	Overall Series (N = 130)									
		3 Months					12 Months				
		Mean (SD) (Range)	MD (SE) CI 95%	t	*p*	d	Mean (SD) (Range)	MD (SE) CI 95%	t	*p*	d
PF	84.7 (24.0) (0.0–100.0)	75.1 (23.1) (10.0–100.0)	−9.6 (2.1) (−13.8–5.4)	−4.5	<0.001	0.4	70.7 (26.5) (0.0–100.0)	−14.0 (2.1) (−18.2–9.8)	−6.6	<0.001	0.6
RP	83.2 (35.2) (0.0–100.0)	38.8 (44.5) (0.0–100.0)	−44.4 (3.1) (−50.5–38.3)	−14.2	<0.001	1.1	41.1 (42.9) (0.0–100.0)	−42.1 (3.1) (−48.2–36.0)	−13.5	<0.001	1.0
BP	79.0 (27.9) (0.0–100.0)	68.2 (27.3) (0.0–100.0)	−10.8 (2.5) (−15.6–6.0)	−4.4	<0.001	0.4	60.6 (31.1) (0.0–100.0)	−18.4 (2.5) (−23.2–13.6)	−7.5	<0.001	0.6
GH	68.3 (22.3) (0.0–100.0)	63.3 (21.7) (15.0–100.0)	−5.0 (2.0) (−8.9–1.1)	−2.5	0.011	0.2	57.2 (24.3) (5.0–100.0)	−11.1 (1.9) (−15.0–7.2)	−5.6	<0.001	0.5
VT	66.9 (22.1) (0.0–100.0)	56.1 (27.3) (0.0–100.0)	−10.8 (2.0) (−14.6–6.9)	−5.5	<0.001	0.4	54.7 (27.6) (0.0–100.0)	−12.2 (1.9) (−16.0–8.4)	−6.2	<0.001	0.5
SF	90.1 (20.0) (0.0–100.0)	69.7 (29.8) (0.0–100.0)	−20.4 (1.8) (−23.9–16.9)	−11.5	<0.001	0.8	74.5 (29.0) (0.0–100.0)	−15.6 (1.8) (−19.0–12.1)	−8.8	<0.001	0.6
RE	88.6 (30.1) (0.0–100.0)	50.7 (46.5) (0.0–100.0)	−37.9 (2.7) (−43.2–32.6)	−14.2	<0.001	1.0	56.7 (46.7) (0.0–100.0)	−31.9 (2.7) (−37.2–26.6)	−11.9	<0.001	0.8
MH	73.3 (20.1) (0.0–100.0)	68.6 (24.1) (0.0–100.0)	−4.7 (1.8) (−8.2–1.2)	−2.6	0.008	0.2	70.1 (23.8) (4.0–100.0)	−3.2 (1.78) (−6.7–0.3)	−1.8	0.072	0.1

In each box, in the first row the mean is indicated and the standard deviation in brackets; and in the second row, the range in brackets. MD: Means differences. SE: Standard Error. CI 95%: 95% confidence interval for the difference in means. *p*: *p* value. t: *t*-student value. d: Cohen’s d value. PF: Physical functioning. RP: Physical role. BP: Bodily pain. GH: General health. VT: Vitality. SF: Social functioning. RE: Emotional role. MH: Mental health.

**Table 4 ijerph-19-10421-t004:** Analysis of the evolution between 3 and 12 months of the domains of the SF-36 questionnaire.

SF-36 Domains	t	*p*	CI95%	d
PF	−0.594	0.554	−9.99 to 5.38	−0.005
RP	0.717	0.474	−2.82 to 6.02	0.05
BP	−1.87	0.064	−9.89 to 0.28	−0.16
GH	−1.368	0.174	−15.66 to 2.86	0.13
VT	−0.648	0.518	−5.36 to 2.70	−0.06
SF	2.495	0.014	0.89 to 7.78	0.17
RE	2.595	0.011	1.68 to 12.49	0.25
MH	3.215	0.002	2.16 to 9.07	0.26
PCS	2.981	0.003	0.088 to 0.448	0.269
MCS	−1.606	0.111	−0.322 to 0.033	−0.145

PF: Physical functioning. RP: Physical role. BP: Bodily pain. GH: General health. VT: Vitality. SF: Social functioning. RE: Emotional role. MH: Mental health. PCS: Physical component summary score. MCS: Mental component summary score. t: *t*-Student. *p*: Degree of significance of the *t*-Student. CI95%: Confidence interval at 95% for the mean difference between the values of the domains at 3 and 12 months. d: Cohen’s d. Effect size: Qualified as small when d = 0.2–0.3; medium when d = 0.5–0.8, and large when d > 0.8.

## Data Availability

The data are not publicly available due to reasons concerning the privacy of the subjects and since it belongs to an ongoing project.

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
