# Peer review of "Impact of COVID-19 on Health-Related Quality of Life: A Longitudinal Study in a Spanish Clinical Sample"

_ijerph, 2022, doi:10.3390/ijerph191610421_

Round 1

Reviewer 1 Report

Thank you very much for giving me the opportunity to revise the article entitled “Impact of COVID-19 on Health-Related Quality of Life: a longitudinal study in a Spanish clinical sample”. This study entails an advance in the comprehension of the consequences of COVID-19 on Health-Related Quality of Life in the long term. The procedure and the employed methodology are adequate to respond the proposed aims. Furthermore, the discussion and conclusion section are linked to the obtained results. However, I would like to propose some minor revisions. Following, the authors can find my comments.

-In the introduction section, authors should explain more exhaustively the specific aims of the study.

-Data analysis should be described in a deeper manner, separating the employed statistical analysis to respond to each aim of the study (e.g.: differences between general and clinical population, differences between 3 and 12 months after the infection).

-Tables 1 and 2 should include the legend of the data (eg.: mean, standard deviation, percentage, frequency…)

-In the results section, it seems that some charts are not visible in the document due to formatting mistakes. I encourage authors to revise it and include all of the charts referred in the Figure 2.

-Taking into account that the SF-36 allows the calculation of two summary scores: the physical summary component (PCS) and the mental summary component (MCS), authors should consider these dimensions in the results.

-Moreover, in the conclusion section, author should include some proposals of approaches from the clinical setting to prevent Health-Related Quality of Life impairment in this population

Author Response

We are very thankful to the anonymous reviewer, who carefully read the manuscript and provided very useful comments to improve the paper.

Here are our point-by-point responses:

-In the introduction section, authors should explain more exhaustively the specific aims of the study.

Thank you very much for your comment. Following your suggestion, we have replaced the sentence:

Therefore, the aim of this work is to study the long-term health status and HRQoL in a Spanish sample of survivors of bilateral SARS-CoV-2 pneumonia.” 

By the following one:

“Therefore, the objective of this study was to evaluate the impact of COVID-19 disease on HRQoL and its evolution over time (at 3 and 12 months after the infection) in a Spanish sample of survivors of bilateral SARS-CoV-2 pneumonia.”

-Data analysis should be described in a deeper manner, separating the employed statistical analysis to respond to each aim of the study (e.g.: differences between general and clinical population, differences between 3 and 12 months after the infection).

The paragraph of section 2.4. Data analysis of section 2. Material and methods has been changed to the following:

 "Qualitative variables are described as absolute frequency and percentage, and quantitative variables are described as mean, standard deviation, and range. Differences between general and clinical population in HRQoL domains were evaluated employing independent T-Tests analysis. To analyze differences at 3 and 6 months after the infection a paired samples T-Test was performed. To assess the size of the effect, Cohen's d was used, and it was classified as small (d=0.2-0.3), medium (d=0.5-0.8) and large (d>0.8). The authors used SPSS for windows, version 24.0.0.0 (IBM Corp. Released 2016. IBM SPSS Statistics for Windows, Version 24.0. Armonk, NY: IBM Corp) and MedCalc Statistical Software version 19.2.6 (MedCalc Software bv, Ostend, Belgium;2020). The limit of statistical significance was set at ≤ 0.05".

-Tables 1 and 2 should include the legend of the data (eg.: mean, standard deviation, percentage, frequency…).

Thank you very much for your comment. Bases on your suggestion, we have included the legend of the data (Frequency and % for qualitative variables and Mean, SD and Range for quantitative data).

-In the results section, it seems that some charts are not visible in the document due to formatting mistakes. I encourage authors to revise it and include all of the charts referred in the Figure 2.

Thank you very much for your appreciation. It seems to be a mistake of formatting. We have included again all the charts in the document and now are fully visible.

-Taking into account that the SF-36 allows the calculation of two summary scores: the physical summary component (PCS) and the mental summary component (MCS), authors should consider these dimensions in the results.

As there is no information regarding PCS and MCS in the general population, it has been no possible to analyze differences in this regard. However, these dimensions have been included in the analyses about the evolution of HRQoL in our clinical sample. Information in this regard has been included in Figure 2 and Table 4.

-Moreover, in the conclusion section, author should include some proposals of approaches from the clinical setting to prevent Health-Related Quality of Life impairment in this population

Based on your comment, the conclusions section has been modified by the following:

 "Our study confirms that COVID-19 pneumonia causes a deterioration in HRQoL measured by the SF-36 questionnaire and this affectation can persist over time. The main clinical practice guidelines for the follow-up of post-COVID-19 patients (NICE, WHO) with persistent symptoms and sequelae at different levels recommend, with a high degree of evidence, the support of patients with multidisciplinary rehabilitation. It includes giving advice and information on self-management to people with ongoing symptomatic COVID‑19 or post‑COVID‑19 syndrome and use a multidisciplinary approach to guide rehabilitation, including physical, psychological and psychiatric aspects of management. Particular attention should be given to vulnerable people, for example older people and people with complex needs. Additional support may include short-term care packages, advance care planning, and support with social isolation, loneliness, and bereavement, if appropriate.

Our results suggest that such care should be maintained in the long term to minimize the overall impact of the disease on the patient's overall wellbeing and lifestyle.

We consider that more evidence and research is needed to understand the impact of the disease on quality of life in order to design more strategies to improve post-COVID-19 patient care".

Reviewer 2 Report

The manuscript “Impact of COVID-19 on Health-Related Quality of Life: a longitudinal study in a Spanish clinical sample” is of interest to physicians and epidemiologists, and the manuscript is well written. Nevertheless, some methodological issues remain.

Major

1. According to the flowchart, the presented study seems to be a complete-case analysis, where especially the better courses drop out of the follow-up, but this is not completely clear from the manuscript (see point 2 of minor issues). Although the proportion of dropouts is very high, shouldn't methods be used that can handle missing values, i.e. likelihood-based methods or multiple imputations? At the moment, the results are selective for worse courses of the disease.

2. Data analysis: the authors mention “In the case of finding statistically significant differences in the test, this difference was expressed with its standard error and the 95% confidence interval.” Does this mean that only significant results are reported?

3. Please specify what is meant by: “The association between quantitative variables was studied using the t-Student test.”

4. Why are discrete and continuous variables divided into Tables 1 and 2?

5. Please add the number of missing values in all descriptive tables.

6. The comparison of the SF-36 with the normal population does not seem to be adequate. Were the data of the normal population matched for age and sex? If so, this should be described under the methods.

6. How were differences between the study sample and the general population evaluated? Did you access the data of the GP? Please specify in the methods.

7. Wouldn't it be more helpful to use data from patients with pneumonia to assess HrQoL, even if HrQoL was measured with a different instrument (e.g. SF-12, VR-12)?

8. Figure 2 suggests that there are considerable proportions in which some components of SF-36 deteriorate from 3 to 12 month; in others considerable improvement is shown. Did the authors examine if some of the measured covariates are associated with improvement/deterioration?

9. In general results should be discussed with less focus on statistical significance (last paragraph page 7 and methods). It is recommended to focus on the effects and confidence intervals (Wasserstein RL, Schirm AL, Lazar NA. Moving to a world beyond “p< 0.05”. Taylor & Francis; 2019.)

Minor

1. Please introduce abbreviations used in the abstract.

2. Overall, the study design is very well described. However, additions to the flowchart would be helpful to break down the number of dropouts per reason.

3. Figure 2 is too small and parts are missing

Author Response

We would like to thank the reviewer for the exhaustive revision of the manuscript and the useful changes and improvements suggested which will significantly increase the quality of the manuscript. Then, we proceed to respond to each of the comments of the reviewer.

  1. According to the flowchart, the presented study seems to be a complete-case analysis, where especially the better courses drop out of the follow-up, but this is not completely clear from the manuscript (see point 2 of minor issues). Although the proportion of dropouts is very high, shouldn't methods be used that can handle missing values, i.e. likelihood-based methods or multiple imputations? At the moment, the results are selective for worse courses of the disease.

Thank you very much for your comment. As can be seen in Figure 1 corresponding to the Flow chart of patient inclusion, the cases that were excluded were due to:

  • Patients preferred not to follow-up their recovery at the Hospital General Universitario Dr. Balmis, where this study was carried out.
  • Patients who were not interested in the 12-month follow-up because at 3 months they were already satisfied with their recovery
  • Patients who did not want to continue with the review at 12 months, and
  • Patients who could not be contacted at 12 months and, therefore, there is no data from the SF-36 questionnaire at that time point.

The rest of the set of patients (130 subjects) provided all information required for all the variables considered (tables 1 and 2), so there are no missing data.

However, based on the comment of the reviewer, we have to take into account that some of the included patients could be those with worse course of the disease, and this fact has been included as a limitation of the study as follows:

“Moreover, it should be considered that some patients that continue with the follow-up were those were suffering the worse courses of the disease and needed treatment. However, heterogeneous results have been obtained in relation to the evolution of the quality of life, improving in some dimensions and worsening in others.”

  1. Data analysis: the authors mention “In the case of finding statistically significant differences in the test, this difference was expressed with its standard error and the 95% confidence interval.” Does this mean that only significant results are reported?

Thank you for your appreciation. No, as can be seen in the article tables, all the results have been exposed. To avoid confusions, this information has been eliminated.

  1. Please specify what is meant by: “The association between quantitative variables was studied using the t-Student test.”

Based on your comment and from the other reviewer, the data analysis section has been fully reformulated to make clearer the performed analyses. The final information is the following:

"Qualitative variables are described as absolute frequency and percentage, and quantitative variables are described as mean, standard deviation, and range. Differences between general and clinical population in HRQoL domains were evaluated employing independent T-Tests analysis. To analyze differences at 3 and 6 months after the infection a paired samples T-Test was performed. To assess the size of the effect, Cohen's d was used, and it was classified as small (d=0.2-0.3), medium (d=0.5-0.8) and large (d>0.8). The authors used SPSS for windows, version 24.0.0.0 (IBM Corp. Released 2016. IBM SPSS Statistics for Windows, Version 24.0. Armonk, NY: IBM Corp) and MedCalc Statistical Software version 19.2.6 (MedCalc Software bv, Ostend, Belgium;2020). The limit of statistical significance was set at ≤ 0.05".

  1. Why are discrete and continuous variables divided into Tables 1 and 2?

Thank you for the comment. It was decided to separate the discrete variables from the continuous ones in order to favor clarity in the exposition, since they are described differently, the former with frequency and percentage values, and the latter with mean, standard deviation and range.

  1. Please add the number of missing values in all descriptive tables.

As we have indicated previously, the cases included are complete cases in all the variables collected, with no missing values, so there is no number of missing cases in the results presented.

  1. The comparison of the SF-36 with the normal population does not seem to be adequate. Were the data of the normal population matched for age and sex? If so, this should be described under the methods.

In relation to the reviewer's comment, the following sentence has been added in the data analysis section:

“For purposes of comparison between our data and the values described for the general Spanish population [12], it was taken into account that the distribution of age and sex in our series was similar to that of the general reference population”

  1. How were differences between the study sample and the general population evaluated? Did you access the data of the GP? Please specify in the methods.

For the comparison of the scores of the different domains of the questionnaire between the study subjects and the Spanish population, the population values ​​collected in the following article were used "Alonso, J.; Regidor, E.; Barrio, J. Valores poblacionales de referencia de la versión española del Cuestionario de Salud SF-36. Med Clin, 1998, 111, 410-416". This article includes a representative sample of the non-institutionalized adult Spanish population and aims to present the reference population values ​​of the SF-36 questionnaire.
The authors of the article offer the possibility of having all the complete data, for example, for the calculation of percentiles. In our case, the means of the values ​​were exposed and we did not request it.

To compare means, we used the statistical program MedCalc. It is a statistical software package designed for the biomedical sciences and includes basic parametric and non-parametric statistical procedures and graphs such as descriptive statistics, ANOVA, Mann-Whitney test, Wilcoxon test, X2 test, correlation, linear as well as non-linear regression, logistic regression and multivariate statistics.

There are other articles in the literature that use the population data from the SF-36 questionnaire for their comparisons using a methodology similar to ours. For example, " van der Sar-van der Brugge, S.; Talman, S.; Boonman-de Winter, L.; de Mol, M.; Hoefman, E.; van Etten, R. W.; De Backer, I. C. Pulmonary function and health-related quality of life after COVID-19 pneumonia. Respiratory medicine, 2021176, 106272", in this article that also deals with the study of quality of life and covid, the results of the SF-36 were compared with the normative data of a randomized study, national sample of 1742 Dutch adults.

  1. Wouldn't it be more helpful to use data from patients with pneumonia to assess HrQoL, even if HrQoL was measured with a different instrument (e.g. SF-12, VR-12)?

The SF-36 health-related quality of life questionnaire is a validated questionnaire, widely used in epidemiological studies in the literature.

There are several articles that study health-related quality of life in covid-19 disease and use this same instrument [5,6,19]. For this reason, we did not consider using other health-related quality of life measurement instruments, nor a summary version of the SF-36 questionnaire.

  1. Poudel, A.N.; Zhu, S.; Cooper, N.; Roderick, P.; Alwan, N. Impact of Covid-19 on health-related quality of life of patients: A structured review. PLOS ONE, 2021, 10, e0259164.
  2. Sanchez-Ramirez, D.; Normand, K.; Zhaoyun, Y.; Torres Castro, R. Long-Term Impact of COVID-19: A Systematic Review of the Literature and Meta-Analysis. Biomedicines, 2021, 9, 900.
  3. van der Sar-van der Brugge, S.; Talman, S.; Boonman-de Winter, L.; de Mol, M.; Hoefman, E.; van Etten, R. W.; De Backer, I. C. Pulmonary function and health-related quality of life after COVID-19 pneumonia. Respiratory medicine, 2021, 176, 106272.
  4. Figure 2 suggests that there are considerable proportions in which some components of SF-36 deteriorate from 3 to 12 month; in others considerable improvement is shown. Did the authors examine if some of the measured covariates are associated with improvement/deterioration?

Thank you very much for this useful comment. Our work has aimed to describe the findings in relation to health-related quality of life in surviving patients with COVID-19 pneumonia, assessed at two time points, at 3 and 12 months. The association between the findings found and their relationship with covariates that may help explain these findings is not the objective of this study, but this analysis is being carried out and will be the subject of future publications. We do not consider apropirate to included so much information that go beyond the aims of the present study.

  1. In general results should be discussed with less focus on statistical significance (last paragraph page 7 and methods). It is recommended to focus on the effects and confidence intervals (Wasserstein RL, Schirm AL, Lazar NA. Moving to a world beyond “p< 0.05”. Taylor & Francis; 2019.)

Thank you very much for this useful comment and article. The mentioned article offers a modern view on the interpretation of the p value. For this reason, we have modified the wording, trying to avoid phrases such as "statistically significant" or "not significant" and table 4 has been in section 3.3,

Evolution in HRQoL scores in patients with post COVID-19 pneumonia at 3 and 12 months after the contagion  has been changed by the following:… The results of the analysis of the evolution of the participants one year after hospital discharge are shown in Table 4 and Figure 2. Although a trend towards improvement was detected in most domains of the SF-36 questionnaire, the effect size was practically null for the RP, VT, SF, RE, MH, and PF domains, although for the PF domain, the difference in scores at 3 and 6 months showed a relevant increase. The BP and GH domains also showed improvement over the testing period, although the effect size could be described as small. Finally, regarding the summary scores, a decrease in PCS was found over time, with a small effect size.

Minor

  1. Please introduce abbreviations used in the abstract.

In response to reviewer 2, the abbreviations have been explained in the abstract.

  1. Overall, the study design is very well described. However, additions to the flowchart would be helpful to break down the number of dropouts per reason.

Thank you very much for your comment. Figure 1 has been modified and the number of dropouts has been broken down by reason.

  1. Figure 2 is too small and parts are missing

Thank you very much for your appreciation. It seems to be a mistake of formatting. We have included again all the charts in the document and now are fully visible.

Round 2

Reviewer 2 Report

Revision of the manuscript “Impact of COVID-19 on Health-Related Quality of Life: a longitudinal study in a Spanish clinical sample”. For the most parts, the authors have created an excellent revision of the article. Nevertheless, this reviewer sees that two of the issues could not be resolved.

(1) Selective dropout

The revised flowchart shows clearly that dropout has been selective. The study therefore represents a complete case study which is the most severe limitation. The newly added limitation should be extended to denote the phrase “complete cases analysis” which lacks generalizability. In addition, it seems advisable to present not only the baseline characteristics of the complete cases but also those of the entire study population in Tables 1/2. This would provide more transparency for the readers.

(2) HrQoL using SF-36

To this reviewer it is still not clear how comparability with the general population has been safeguarded. The author response: “For purposes of comparison between our data and the values described for the general Spanish population [12], it was taken into account that the distribution of age and sex in our series was similar to that of the general reference population” is too vague. What means similar? Due to a study population of patients suffering from Covid-19 infection, which imposed more burden to the elderly with chronic comorbidities in the beginning of this pandemic, a similarity with the GP is not comprehensible and should be shown in this study.

Author Response

(1) Selective dropout: The revised flowchart shows clearly that dropout has been selective. The study therefore represents a complete case study which is the most severe limitation. The newly added limitation should be extended to denote the phrase “complete cases analysis” which lacks generalizability. In addition, it seems advisable to present not only the baseline characteristics of the complete cases but also those of the entire study population in Tables 1/2. This would provide more transparency for the readers.

Thank you very much for your comment. To facilitate data transparency, Tables 1 and 2 have been modified, and all available data from study participants have been included.

In addition, the limitations of the study have been modified by the following text:

"Moreover, the selective dropout of participants should be considered, mainly due to lack of motivation, which has generated that some patients who continue with follow-up were those who presented the worst evolution of the disease and needed treatment. However, heterogeneous results have been obtained in relation to the evolution of quality of life, improving in some dimensions and worsening in others".

(2) HrQoL using SF-36

To this reviewer it is still not clear how comparability with the general population has been safeguarded. The author response: “For purposes of comparison between our data and the values described for the general Spanish population [12], it was taken into account that the distribution of age and sex in our series was similar to that of the general reference population” is too vagueWhat means similar? Due to a study population of patients suffering from Covid-19 infection, which imposed more burden to the elderly with chronic comorbidities in the beginning of this pandemic, a similarity with the GP is not comprehensible and should be shown in this study.

Thank you for your comment. Taking into account that we don not have had the possibility to match both samples by age and gender, we have included this fact as a limitation of the study and something to consider in future studies as follows:

“Furthermore, it was not possible to specifically match by sex and age the comparison groups (general population and clinical sample) due to the data from general population was extracted from a previously published paper. This fact could limit the generalization of the obtained results and entails taking with caution the results obtained. Future studies should address this issue including a control group from the general population specifically matched by these sociodemographic dimensions to analyze the possible differences in HRQoL.”
